# Performance of a Real Time PCR for *Pneumocystis jirovecii* Identification in Induced Sputum of AIDS Patients: Differentiation between Pneumonia and Colonization

**DOI:** 10.3390/jof8030222

**Published:** 2022-02-24

**Authors:** Oscar José Chagas, Priscila Paiva Nagatomo, Vera Lucia Pereira-Chioccola, Ricardo Gava, Renata Buccheri, Gilda Maria Barbaro Del Negro, Gil Benard

**Affiliations:** 1Laboratório de Investigação Médica em Micologia LIM53, Instituto de Medicina Tropical e Hospital das Clínicas, Faculdade de Medicina da Universidade de São Paulo, São Paulo 05403-000, Brazil; gildadelnegro@gmail.com (G.M.B.D.N.); bengil60@gmail.com (G.B.); 2Laboratório de Biologia Molecular de Parasitas e Fungos do Centro de Parasitologia e Micologia, Instituto Adolfo Lutz, São Paulo 01246-000, Brazil; pchioccola@gmail.com (V.L.P.-C.); gava44@gmail.com (R.G.); 3Instituto de Infectologia Emílio Ribas, São Paulo 01246-900, Brazil; renatabuccheri@gmail.com

**Keywords:** *Pneumocystis jirovecii*, HIV/AIDS, molecular diagnosis, real time PCR, 1,3-β-D-glucan

## Abstract

*Pneumocystis jirovecii* pneumonia (PcP) remains an important cause of morbimortality worldwide and a diagnostic challenge. Conventional methods have low accuracy, hardly discriminating colonization from infection, while some new high-cost or broncho-alveolar lavage-based methods have limited usefulness in developing countries. Quantitative PCR (qPCR) tests may overcome these limitations due to their high accuracy, possibility of automation, and decreasing cost. We evaluated an in-house qPCR targeting the fungus mtSSU gene using induced sputum. Sensitivity of the assay (ten target gene copies/assay) was determined using recombinant plasmids. We prospectively studied 86 AIDS patients with subacute respiratory symptoms in whom PcP was suspected. qPCR results were determined as quantification cycles (Cq) and compared with a qualitative PCR performed in the same IS, serum 1,3-β-D-Glucan assay, and a clinical/laboratory/radiology index for PcP. The qPCR clustered the patients in three groups: 32 with Cq ≤ 31 (qPCR+), 45 with Cq ≥ 33 (qPCR-), and nine with Cq between 31-33 (intermediary), which, combined with the other three analyses, enabled us to classify the groups as having PcP, not *P. jirovecii*-infected, and *P. jirovecii*-colonized, respectively. This molecular assay may contribute to improve PcP management, avoiding unnecessary treatments, and our knowledge of the natural history of this infection.

## 1. Introduction

HIV is still an important cause of death across the world, [1] and *Pneumocystis* pneumonia (PcP) remains among the leading pulmonary opportunistic infections (OI) in several developing and developed countries [2,3,4,5,6,7]. However, data of PcP incidence are scarce mainly in developing countries, and the largest studies of invasive fungal infections (IFI) seldom included PcP due to the difficulties related to its diagnosis [5,8,9,10,11,12,13]. In Brazil, it has been reported to range from 3.2% to 47.1% [14,15,16,17,18,19,20], and it is estimated that PcP accounts worldwide for almost 400,000 cases/year, with 200,000 deaths/year, mainly in developing countries [21].

PcP diagnosis remains a challenge, owing to several issues such as the absence of culture systems for *P. jirovecii*, low accuracy of the conventional diagnostic methods, and difficulties in sample collection. Methods based on visualization of the fungus with conventional staining have low sensitivity, requiring broncho-alveolar lavage fluid (BALF) samples for better performance. The commercially available immunofluorescence kits present higher sensitivity; however, they are not only generally expensive, but also require BALF samples for better results [22,23,24,25,26,27]. However, bronchoscopy is an invasive and expensive procedure, thus not easily available in public health services of developing countries [28,29].

There are many reports on PCR assays for PcP diagnosis, with greater overall sensitivity. However, the use of different samples (oral wash, sputum, induced sputum, BALF), in different settings (HIV, other immunosuppressed conditions), with different molecular techniques and gene target primers, have hampered the standardization and more widespread application of this technique [30,31,32,33,34,35,36,37,38,39,40,41]. Furthermore, a major concern arises as molecular methods detect a lower number of fungi than direct visualization methods, which could be associated with just colonization instead of infection [42,43,44,45,46,47,48,49,50,51]. Quantitative PCR studies were designed with the aim of discriminating infected from colonized patients but did not achieve this goal due to the lack of reliable gold standard diagnostic techniques. Several of these studies described “intermediary” or “gray” zones, corresponding to cases with inconclusive results, and leading to classifications such as possible, probable, or suspected PcP [33,36,37,52,53,54,55,56].

1,3-β-D-glucan (BDG) dosage has also been applied in the diagnosis of PcP. However, due to its limitation of yielding non-specific results, the cutoff values for diagnosing PcP may be higher than usually recommended by the manufacturer, and the test may require confirmation through association with PCR assays [57,58,59]. Furthermore, due to its high cost, is also rarely available in developing countries.

This scenario is further complicated by the fact that AIDS patients often present concurrent pulmonary OIs, which lead to clinical and radiological features that overlap with those of PcP [60]. In these cases, the identification of *P. jirovecii* in respiratory samples can indicate either PcP or colonization [54]. Overall, these considerations stress the need for more suitable diagnostic methods aiming at improving the approach to patients with PcP and the knowledge of its epidemiology.

We thus carried out a prospective study in aids patients with suspected PcP of an in-house real-time PCR assay (qPCR) to quantitate *P. jirovecii* in induced sputum (IS). We took into account the clinical characteristics of the patients, and compared our qPCR with the BDG assay and a conventional qualitative PCR assay (cPCR). IS has the advantage over “spontaneous” sputum because it retrieves more frequent lung samples from the upper bronchial tree and, compared with BALF sampling, is easier to perform and can be done at the bedside, since it does not require an invasive procedure [28,30,40,61,62,63,64,65]. We show here that our qPCR, standardized for IS samples, presents a high performance when applied to AIDS patients suspected of having PcP, allowing the discrimination between infection and colonization.

## 2. Materials and Methods

### 2.1. Study Design and Patients

This was a prospective study that enrolled AIDS patients admitted between December 2017 and February 2020 at the emergency unit of the IIER, who were initially suspected of having PcP according to the following criteria: the presence of subacute cough and dyspnea (≥7 days), a current TCD4 count <250 cells/mm^3^, and poor compliance to or not on ART and PcP prophylaxis. IS was collected before starting treatment for PcP (or with up to one dose), through inhalation of hypersaline solution (3% to 5% of NaCl), for 15–20 min, collected in a sterile container and stored at 4 °C until DNA extraction up to the next day. The study was approved by the IIER ethics committee (protocol 06/2016), and all patients signed an informed consent form.

In addition to the admission criteria, we defined specific features that enhanced the likelihood of PcP: (a) presence of dry cough (as opposed to productive cough); (b) interstitial infiltrates and/or ground-glass opacities on X-ray or CT-scan, respectively; (c) CD4 counts inferior to 100 cells/mm^3^; (d) LDH values greater than 350 UI/L; (e) O₂ saturation < 94.0% by pulse oximetry at room atmosphere; (f) positive clinical response to PcP treatment with cotrimoxazole or alternative regimens; (g) absence of treatment for other concurrent pulmonary diagnoses (most commonly tuberculosis [TB], cytomegalovirus [CMV], and community-acquired bacterial pneumonia [CABP]). In addition, the result of direct colorimetric examination performed in IS samples from a subset of patients was also considered: it was positive in eight patients out of the first 55 patients enrolled (it was not available for the subsequent patients for technical reasons). Those with at least three of the above features were considered, based on our clinical expertise, as having enhanced likelihood for PcP (high clinical index), while those with less than three features were considered as having a low likelihood (low clinical index). This analysis was done blinded to the laboratory results by one of the authors (G.B.).

### 2.2. DNA Extractions and PCR Assays

One mL of the IS samples was used for DNA extractions, which were carried out with a QIAamp DNA mini kit (Qiagen, Hilden, Germany). Sputum samples were previously dissolved *v*/*v* in PBS pH 7.2, mixed, and shaken in a vortex. Then, tubes were centrifuged (2.500× *g*/10 min), and cell pellets were digested at 56 °C with 20 μL of proteinase K 20 mg/mL in 180 μL of AL buffer (Qiagen) until complete tissue digestion. The remaining steps were performed according to the manufacturer’s kit protocol. DNA samples concentration and purity were estimated by UV spectrophotometry (Nanodrop 1000, Thermo Scientific) at 260 nm and 280 nm.

The cPCR was previously established based on Khan et al., 1999 [65]. Amplifications were performed with primers pAZ102-E (5′-GATGGCTGTTTCCAAGCCCA-3′) and pAZ102-H (5′-GTGTACGTTGCAAAGTACTC-3′) targeting the large subunit mitochondrial RNA (mtLSU) gene from *P. jirovecii*. PCR products (345 bp) were electrophoresed in 2% agarose gels, stained with ethidium bromide, and visualized under UV transilluminator apparatus (MiniBis Pro, DNR, Bio Imaging Systems, Israel) [65].

The qPCR assay was performed with primers previously described, targeting the *P. jirovecii* mitochondrial small subunit (mtSSU) gene, and resulting in 76 bp amplicon [66]. A total of 20 microliters of reaction mix containing 10 μL of 2× Power SYBR Green PCR Master Mix (Applied Biosystems, Foster City, CA, USA), 0.25 μM of each primer, and 5 µL of DNA clinical samples were submitted to cycling conditions consisting of an initial denaturation step (95 °C, 5 min), followed by 40 cycles at 95°C for 30 s, 62 °C for 30 s, and 72 °C for 30 s, and a final dissociation step at 62–95 °C with a heating rate of 0.3 °C/s, in the Biofast 7500 PCR System (Applied Biosystems).

The standard curves were constructed using recombinant plasmids obtained from the 76-bp PCR product cloned with the CloneJET PCR Cloning Kit (Thermo Fisher Scientific, Wilmington, DE, USA), according to the manufacturer’s instructions. To establish the limit of detection (LOD), qPCR assays were performed with serial 10-fold dilutions of the recombinant plasmids (1 × 10^6^ to 1 × 10^0^ copies/µL) in 20 independent experiments. The LOD was defined as the lowest number of copies detected in ≥ 95% of the experiments [67].

The qPCR assay specificity was also evaluated by testing the P. jirovecii primers with DNA samples from *Paracoccidioides brasiliensis*, *Histoplasma capsulatum*, *Aspergillus* sp., *Rhizopus* sp., *Fusarium* sp., *Candida* sp., *M. tuberculosis*, *A. baumannii*, *P. aeruginosa*, *S. aureus*, *S. pneumoniae*, and *K. pneumoniae*.

All experiments were run in duplicate, and negative controls and standard curves were included in each assay. Since quantification of *P. jirovecii* as copies/mL of the IS sample is not always reliable due to patients’ inherent variations in the quality of the sputum [64], we opted to express the results as quantification cycle (Cq), with higher Cq values associated with lower fungal loads.

### 2.3. 1,3-β-D-Glucan Assay

Serum samples collected simultaneously to the IS were tested with the Fungitell^®^ assay (Associates of Cape Cod, East Falmouth, MA, USA) for BDG measurement according to the manufacturer’s instructions. The positivity cutoff was established at ≥ 150 pg/mL in our institution.

### 2.4. Statistical Analysis

Qualitative data were compared using Pearson’s Chi-square test and quantitative data was compared using ANOVA adjusted for white, and post-hoc analyses with Tukey’s test. Data analyses were performed using Integrated Development for R (RStudio, PBC, Boston, MA, USA) [68]. Statistical significance was set at < 0.05.

## 3. Results

### 3.1. Efficiency and Analytical Sensitivity of the qPCR Assay

The standard curves showed linearity with a high correlation coefficient (R2 = 0.998) and an efficiency of 98%. The LOD of the qPCR assay achieved 10 copies of the recombinant plasmids/assay (detected in 100% of 20 independent experiments), corresponding to a mean Cq value of 32.01 ± 0.95 (CV = 2.98%), and a temperature melting of 76 °C ± 1 °C. The qPCR assay showed 100% analytical specificity, as no amplification signal was detected when all DNA samples from other fungal and bacterial strains were tested.

### 3.2. Patients Enrolled

A total of 97 PWH admitted to the emergency unit of the IIER with respiratory manifestations suggestive of PcP were enrolled. Of these, eight patients were excluded for being transferred to another health service in the first 24 h, and three for not being able to provide IS. Therefore, 86 patients underwent the radiology and laboratory evaluation prescribed by the attending physician. In addition, each patient provided on admission IS for DNA extraction and a blood sample for BDG quantification. Both were carried out later only for the study purpose: the results were not available to the attending physician. Both PCR assays were performed on the same DNA obtained from the IS samples as described in the Section 2.

### 3.3. qPCR Assay vs. Patients’ Clinical, Laboratory and Radiology Characteristics

Of the 86 patients enrolled in the study, the qPCR assay resulted positive (Cq < 33) in 41. We then analyzed the patients’ distribution according to their Cq values and high or low clinical index for PcP. Figure 1 shows that the qPCR Cq values clustered the patients in three distinct groups, one (positive) with a Cq ≤ 31 (n = 32), one (negative) with Cq ≥ 33 (n = 45), and the third group with a positive Cq that was greater than 31 and inferior to 33 (n = 9), provisionally named “intermediary” group. Overall, 29 of the 32 qPCR-positive patients presented a high clinical index for PcP. Of the three patients with a low clinical index for PcP, one patient (#53), who was afflicted with several life-threatening OIs, including neurocryptococcosis, died during the hospitalization, and had a highly positive BDG value (>500 pg/mL), all of which pointed to PcP. One patient (#14) had a negative BDG test, and was treated solely for CABP with an improvement of the respiratory manifestation, pointing to *P. jirovecii* colonization only. It is of note that his/her Cq (31.0) was at the upper Cq limit of the qPCR-positive patients’ cluster. The third one (#73, Cq: 31.5) had a positive BDG test (310 pg/mL), but improved without specific treatment for PcP, which suggests colonization, although PcP could not be ruled out.

On the other hand, among the 45 qPCR-negative patients, 33 were accordingly classified as unlikely PcP based on their low clinical index. The remaining 12 qPCR-negatives, with a high clinical index for PcP, were also diagnosed with a concurrent pulmonary infection (mainly TB, CABP, or presumed CMV pneumonitis) that could partly mimic the manifestations of PcP. They were all simultaneously treated for the concurrent pulmonary diseases and PcP, showing clinical improvement. However, 10 of these 12 patients had a negative BDG test, which, combined with the qPCR-negative (and negative cPCR, see below) result, reinforced the unlikeliness of PcP. Of the two with a positive BDG test, one patient (#52), who had a BDG value (158 pg/mL) very close to our cutoff value, was treated for both PcP and CABP with clinical improvement. We considered the patient as not being infected with *P. jirovecii*, since the BDG values in patients with positive (Cq ≤ 31) qPCR and high clinical index for PcP were almost invariably higher than 250 pg/mL. The other patient (#37) with a high clinical index for PcP also had a highly elevated BDG value (>500 pg/mL), and can therefore represent false-negative qPCR and cPCR results.

Interestingly, of the nine patients in the intermediary qPCR group (31 > Cq < 33), seven had both low clinical index for PcP and a negative BDG test, which, combined with the high Ct values, led to the diagnosis of colonization. One patient had a high clinical index (#40) but undetectable BDG, and was treated for bacterial pneumonia alone, which resolved the respiratory manifestations, leading to the diagnosis of *P. jirovecii* colonization. The last patient (#100) had a low clinical index for PcP but a high BDG value (>500 pg/mL). However, his/her clinical follow-up also ruled out PcP: the patient was treated with levofloxacin alone, resulting in the symptoms’ resolution.

### 3.4. qPCR vs. cPCR

The cPCR was positive in all 32 positive (Cq ≤ 31) qPCR patients and negative in all 45 qPCR-negative patients, reinforcing the accuracy of molecular methods. In contrast, there was a pronounced disagreement in the intermediary group (Figure 2). This was because six of nine patients in this group were non-reactive on the cPCR assay. Based on the fact that cPCR is less sensitive than qPCR, the six discordant results suggest that these patients had a lower fungal load, consistent with a diagnosis of colonization, as already highlighted above. The remaining three patients with cPCR and qPCR-positive results were in fact patients colonized with *P. jirovecii*: the patients #40 (Cq: 32.5) and #100 (Cq: 31.5) detailed above, and patient #50 (Cq: 32.7), with a low clinical index for PcP and negative BDG, resulting in the resolution of the respiratory symptoms without treatment for PcP. Thus, the nine patients with high Cq values (between 31 and 33) were probably colonized. Overall, the qPCR agreed with the cPCR, except for those patients with a positive qPCR with high Cq values.

### 3.5. qPCR vs. BDG

We also compared the qPCR with BDG results (Figure 3). The BDG assay has high sensitivity but low specificity, and can yield false-positive results due to several interferences [66]. Of the 32 qPCR-positive patients, 30 were also strongly BDG-positive, thus directing to the diagnosis of PcP. Of the two qPCR-positive/BDG-negative patients, one of them (#49) had a high clinical index for PcP, but a definite diagnosis could not be established, since the patient received treatment for both CABP and PcP, evolving to the resolution of the symptoms. The other patient (#14) was already detailed above, and was diagnosed as colonization.

In the same way, of the 45 qPCR-negative patients, 40 were concurrently BDG-negative, ruling out PcP infection. Among the five with a discordant positive BDG value, there were two patients who were also detailed above: one of them (#52) with colonization, and the other (#37) with PcP and a probable false-negative qPCR result. The other three had positive BDG tests, but two patients (#42 and #93) had low clinical indices, were treated for an infection other than PcP pulmonary OI, and presented the resolution of the pulmonary symptoms, ruling out PcP. The remaining patient (#30) had a high clinical index, and was treated for disseminated TB and CABP, but died six days after admission; thus, as frequently happens in cases such as this, the diagnosis of PcP could neither be established nor ruled out.

Finally, as mentioned earlier, among the nine patients in the intermediary group, eight were BDG-negative, and the single one (#100) with a positive BDG test was also diagnosed as colonization.

Overall, the BDG values were above the cutoff in 94% of the qPCR+ patients vs. 11% in both the qPCR- and colonization groups (*p* < 0.001). Of note, in the patients with a positive qPCR, there was a significant inverse correlation between the Cq and BDG values (*p* < 0.001 and r^2^ = −0.55).

### 3.6. Clinical, Laboratory and Radiology Characteristics of the Three Patients’ Groups Defined by the qPCR

The three groups showed important clinical, laboratory, and radiology differences, with the Cq ≤ 31 qPCR+ group presenting significantly more features indicative of PcP and advanced AIDS than the other two groups, as depicted in Table 1. This group presented more frequently a dry cough, while the productive cough predominated in the qPCR- and intermediary groups. Consistent with this, the qPCR+ group presented more frequently tachypnea and ground-glass opacities/diffuse infiltrates on pulmonary imaging, lower TCD4 counts and O2 saturation levels, but higher DHL values and HIV viral load. On the other hand, the three groups did not significantly differ regarding age, sex, days since the onset of symptoms, fever, and presence of other pulmonary OIs.

## 4. Discussion

We show here the results of a specific real-time PCR developed to quantify *P. jirovecii* in IS of AIDS patients suspected of having PcP who were prospectively admitted to our emergency unit. We focused on IS instead of BALF samples since bronchoscopy is not easily accessible in public health services of developing countries, such as in our institution. We found that the Cq values discriminated these patients into three groups. We then analyzed these groups, taking into account the clinical, radiology, and laboratory parameters usually associated with PcP, as well as the results of a serum BDG assay, and a qualitative PCR. With these additional parameters, we were able to characterize the three groups as a group having PcP, a group not infected with *P. jirovecii*, and a smaller group of patients likely colonized.

Of the 86 patients enrolled, 32 (37.2%) were qPCR+ with a Cq ≤ 31, 45 (52.3%) were qPCR-, and nine (10.5%) presented an intermediary Cq (between 31 and 33). Compared with the other two groups, the qPCR+ group displayed more frequently a set of clinical-laboratory parameters suggestive of PcP, which included dry cough, interstitial diffuse infiltrates on X-ray or “ground glass” on a CT scan, augmented DHL values, and decreased O2 saturation, together with greater immunosuppression (the lowest TCD4 count: median of 40 cells/mm^3^). A total of 90% of them were initially prescribed with anti-PcP treatment, a proportion significantly higher than the other groups. In fact, the qPCR+ group presented a median score of 5, significantly higher than the qPCR- (median: 2) and colonized groups (median: 1) (Table 1). All of these features strongly point to the Cq ≤ 31 as being a good marker of PcP. This was further corroborated by the high BDG values of this group, with 30 of the 32 patients presenting a high serum BDG level. In fact, of the 32 patients, only one was diagnosed as experiencing colonization (#14), and in one of them, the diagnosis remained inconclusive (#73). Of note, the qPCR+ group included all eight patients with a positive direct colorimetric test.

There are many reports on PCR assays for diagnosing PcP in patients with a variety of immunosuppressed conditions. Although they, in general, reported a good performance, they were not frequently included in the routine clinical practice. One of the most commonly reported limitations in these studies is the lack of a reliable gold standard method, such that the rate of true false-negative results is difficult to estimate [69]. In addition, several studies describe an ill-defined gray zone [41,53,54,56]. We tried to partially overcome this difficulty by gathering the qPCR, a conventional qualitative PCR performed in the same IS samples, the patients’ clinical, laboratory, and radiology evaluation and follow-up of the patients, and a BDG test, as well as by enrolling a relatively homogenous cohort of AIDS patients in whom PcP was suspected.

Another point that we believe contributed to the good performance of our qPCR assay was the choice of the target gene. We employed primers from the mtSSU ribosomal gene, which has been demonstrated to be present with a higher number of copies compared to the mtLSU gene. In addition, the number of mtSSU gene copies neither varies with *P. jirovecii* physiological state nor, in contrast to other genes, decreases with low fungal loads [40,70]. This likely explains the better sensitivity of our qPCR compared with the cPCR in this study and with other qPCR assays, which employ as target primers from the mtLSU gene or other single-copy genes, allowing for the identification of just colonized cases [33,37,69,71,72]. Of note, our cPCR agreed with the qPCR, except for those patients with high Cq values (31 > Cq < 33). Although this qualitative technique also presented a relatively good performance, it was not able to differentiate colonized from infected patients. Colonization is a major issue in AIDS patients, as it has been reported to occur in up to 68% of them [41,42,43,44,45,46,47,48,49,50]. Some authors recommend that patients with colonization should be closely monitored [52]. In addition, colonized patients as the source of in-hospital transmission of *P. jirovecii* and PcP outbreaks have been reported [51,73].

With regard to specificity, we found that our qPCR negative group, with one exception and a case whose diagnosis was inconclusive, was composed of patients not infected with *P. jirovecii*. All 45 qPCR- patients presented a concomitant negative cPCR, while the BGD test was negative in 40. Of the five patients with a positive BDG test, in three of them the presence of *P. jirovecii* was clinically ruled out, in one it was inconclusive, and in the other, the qPCR failed in diagnosing PcP.

The role of the BDG assay in the diagnosis of PcP has been intensely discussed: its major asset is its high negative predictive value, but its high sensitivity and lack of specificity due to factors that interfere with the tests [66,74,75,76,77] can lead to false-positive results. Based on this, it was recommended as a screening test and that higher cutoff values should be used for the diagnosis of PcP, up to 400 pg/mL in AIDS or non-HIV infected patients [57,78,79]. Indeed, in our study, the group of patients having PcP presented mean BDG values of 441 pg/mL (Table 1). Unfortunately, due to its high cost, this test is still not accessible for routine screening in most public or private laboratories in Brazil.

On the other hand, 12 of the 45 qPCR- patients presented a high clinical index for PcP. In all of these patients, pulmonary OIs other than PcP were diagnosed that could account for the respiratory manifestations, in agreement with the fact that concurrent pulmonary OIs are a rather common feature in AIDS patients. In total, nine of these patients had a negative BDG test, reassuring that PcP was not the cause of respiratory symptoms, and in one of them, the test was close to the cutoff value and considered by the clinical staff as not having PcP. The remaining two correspond to the false-negative (#37) and inconclusive diagnosis (#30) patients.

The false-negative qPCR result in patient #37 (and eventually patient #30) points to an important but overlooked aspect of IS-based diagnostic methods that can lead to false-negative results. Our view is that the qPCR-negative result was possibly correct in that the fungus was truly absent from the IS sample, as we suspect was due to non-optimal IS collection. IS a simple procedure, but requires a standardized protocol, as well as the patient’s compliance, since hypertonic saline inhalation provokes mucosal irritation. The procedure yields specimens from the upper bronchial tree initially, and only after 15–20 min, samples from more peripheral airways [28]. Proper IS collection requires monitoring.

The other issue is that the patients with a qPCR-positive test with higher Cq values, forming a cluster of results usually called the gray or intermediary zone. Our analyses of the nine patients who fell in this category showed that most of them had a low clinical index for PcP, a negative BDG test, and a negative cPCR in the same DNA sample, and were diagnosed as colonized patients. The two who did not strictly follow this rule had at least a low clinical index for PcP or a negative BDG test, but a clinical follow-up that ruled out PcP.

Quantitative PCR assays are faster and easier to perform as an automated platform when compared with conventional PCR methods [80]. We additionally chose to employ the Sybr Green fluorescent dye instead of hydrolysis probes, further reducing its cost. Moreover, the possibility of quantification allowed the discrimination between infected and colonized patients. Based on our clinical expertise and the qPCR results, we propose that our colonized group (31 > Cq < 33) should not start immediately on anti-PcP treatment, but be closely monitored since it is known that colonization can eventually lead to the development of PcP [51]. On the other hand, a negative qPCR ruled out PcP provided that the IS sample was properly collected [81]. Taking into account these features, the technique may enable better clinical management, avoiding unnecessary treatments, and a better knowledge of the still poorly known epidemiology of *P. jirovecii* infections.

This study has some limitations. Microscopic examinations were not available for most patients enrolled in our study due to the lack of specialized personnel. These methods usually require laboratory technicians with specific training. They are usually employed as the gold standard, especially when BALF samples are used; with IS, their sensitivity tends to decrease [30]. As a result, the construction of true positive control groups is frequently a major concern in PcP diagnosis studies, hampering reliable calculations of sensitivity, specificity, and predictive values, as well as validation studies. In this regard, the combined utilization of a clinical score was very helpful in defining patients’ classification. Finally, our results are from a single-center investigation; further multi-center studies to validate our qPCR assay are warranted.

In conclusion, this study shows that our in-house qPCR platform targeting the mtSSU gene from IS samples was able to discriminate three groups among AIDS patients according to the Cq values: one with PcP, one without PcP, and a third one with colonized patients. In this regard, this qPCR may represent an alternative method to microscopic examination in the diagnosis of PcP, as well as improve our knowledge of the natural history of this infection.

## Figures and Tables

**Figure 1 jof-08-00222-f001:**
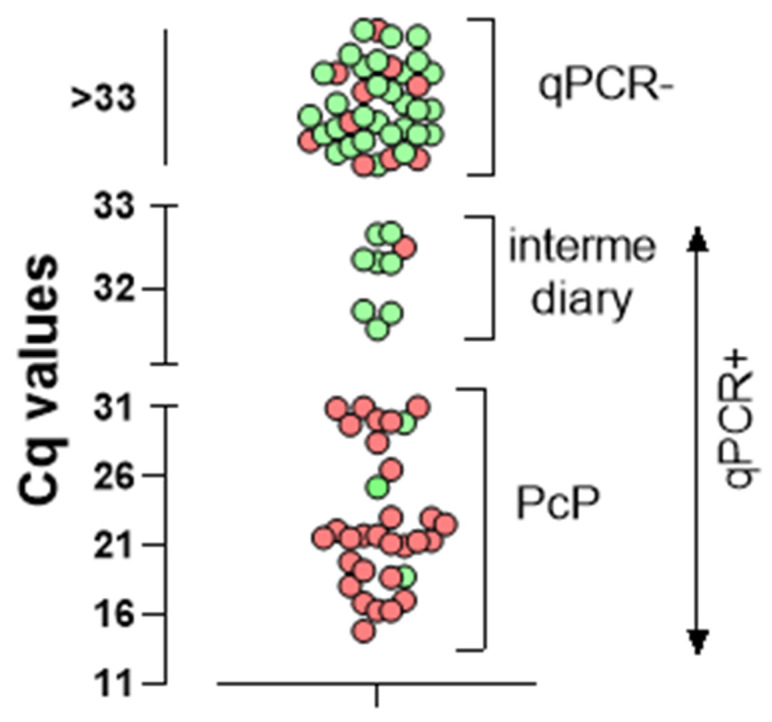
Distribution of the Cq results of the qPCR performed with IS of patients with AIDS and suspected *Pneumocystis* pneumonia (PcP) according to the low (green) or high (red) clinical index for PcP. Each symbol represents one patient. PcP (n = 32): positive qPCR with low Cq values indicating high fungal load. Intermediary group (n = 9): positive qPCR with high Cq values indicating low fungal load. qPCR-: n = 45.

**Figure 2 jof-08-00222-f002:**
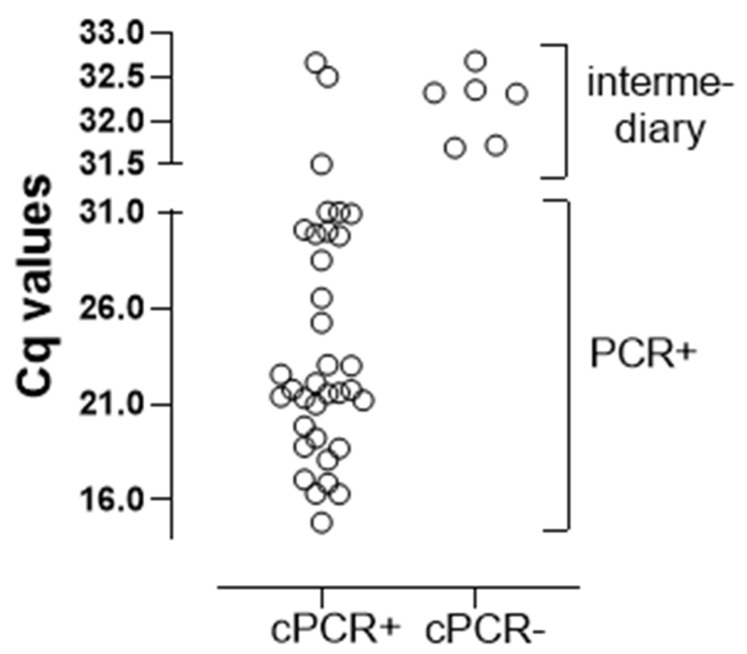
Comparison of the Cq values of the positive qPCR (n = 41) with the results of the cPCR performed in IS of patients with AIDS and suspected PcP. Each symbol represents one patient.

**Figure 3 jof-08-00222-f003:**
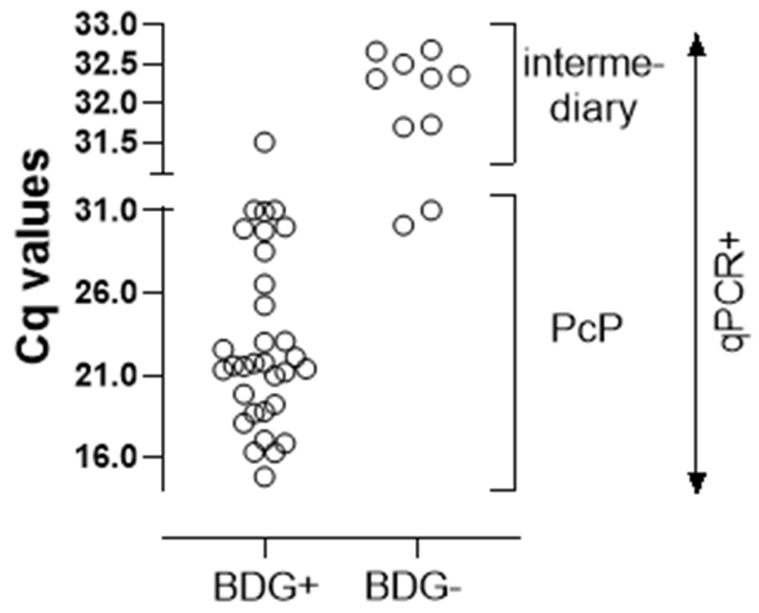
Comparison of the Cq values of the positive qPCR (n = 41) performed in IS with the serum 1,3-beta-d-glucan assay of patients with AIDS and suspected PcP. Each symbol represents one patient.

**Table 1 jof-08-00222-t001:** Clinical, radiological and laboratorial data from the three groups of patients according to their Cq values.

	A Cq ≤ 31 (n = 32)	B 31 > Cq < 33 (n = 9)	C Cq ≥ 33 (n = 45)	*p* Value
Male/Female	27/5	6/3	33/12	0.4
Age (years)	38 ± 2.3	34 ± 3.7	41 ± 3.9	0.15
Dry cough	60%	11%	25%	A × B: 0.02 A × C: 0.004
Productive cough	40%	89%	75%	A × B: 0.02 A × C: 0.004
Fever	78%	77%	73%	0.88
PcP treatment	90%	11%	28%	A × B: <0.001 A × C: <0.001
Tuberculosis *	18%	11%	37%	0.09
Presumed pulmonary CMV *	25%	0%	11%	0.052
Presumed CABP *	68%	33%	42%	A × B: 0.054 A × C: 0.027
Respiratory frequency > 24 bim	84%	44%	57%	A × B: 0.014 A × C: 0.013
Interstitial infiltrate on X-ray	93%	66%	61%	A × B: 0.028 A × C: 0.001
Ground-grass on CT-scan	93%	42%	38%	A × B: 0.001 A × C: <0.001
beta-D-glucan > 150 pg/mL	93%	11%	11%	A × B: <0.001 A × C: <0.001
Days with symptoms	43 ± 9	41 ± 15	43 ± 15	0.98
O2 saturation (%)	88 ± 1.5	91 ± 2.4	93 ± 2.5	A × C: 0.01
Lactic dehydrogenase (U/L)	488 ± 49	219 ± 78	324 ± 81	A × B: 0.004 A × C: 0.004
TCD4 cells/mm^3^	40 ± 15	86 ± 28	75 ± 28	A × C: 0.01 A × B: 0.07
HIV viral load (copies/mL)	780,289 ± 207,058	117,728 ± 386,509	304,545 ± 394,559	A × C: 0.02 A × B: 0.07
BDG (pg/mL)	441 ± 29	76.3 ± 46	62.2 ± 48	A × B: <0.001 A × C: <0.001
PcP clinical index	5 (4 − 5)	1 (1 − 2)	2 (1 − 3)	A × B: <0.001 A × C: <0.001

CABP: community acquired bacterial pneumonia; bim: breath incursions per minute, PcP: *Pneumocystis* pneumonia.; Continuous data expressed as mean ± SE or median and interquartile range and analyzed with ANOVA adjusted for white and post-hoc analyses with Tukey’s test.; Non-continuous data expressed as percentage and analyzed with Pearson’s chi-square; * Some patients had two or more concurrent pulmonary OIs.

## Data Availability

Data available on request from the corresponding author due to ethical restrictions regarding patients’ data.

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
