# Peer review of "Performance of a Real Time PCR for Pneumocystis jirovecii Identification in Induced Sputum of AIDS Patients: Differentiation between Pneumonia and Colonization"

_jof, 2022, doi:10.3390/jof8030222_

Round 1

Reviewer 1 Report

The authors have replied to my comments appropriately.

Author Response

Thanks for the comments!

Reviewer 2 Report

 Some points from my review were not addressed in the new manuscript, and I was not able to see the answer from the authors so that I could understand their point of view. The points that were not solved are listed below:

* Inclusion criteria should include recent Pneumocystis prophylaxis, with a statistical analysis to discard a potential confounding variable (previously in Line 107, now line 105).

* According to the CDC Revised Surveillance Case Definition for HIV Infection (2014), for stage 3 disease, the CD4 T cell count should be <200 cells/mm3; please provide your reference on <250 cells  (previously in Line 108, now line 106).

* If sputum samples are variable, results might not be trusted. Although you addressed this issue in the Discussion section, there should be a way to minimize the variation in the sample collection, for example, quantifying total protein content, or performing a statistical analysis that show no difference between samples  (Line 162).

* Could you explain why you chose 150 pg/mL as the B-glucan cutoff? Other studies have a higher cutoff value (see doi: 10.2478/raon-2020-0028) (previously in Line 109, now line 108).

Author Response

Reviewer 1 – first submission (now reviewer 2 - resubmission)

            We apologize for the confusion. We did respond to the comments in an attachment to the first resubmission. However, as it turned out to be a new submission, probably our response was not made available to your consideration. We provide it again below, and we hope we appropriately addressed the points raised by you.

We would like to thank the reviewer for the careful reading of our article and for the nice comments.

Comment: Line 107 – now line 100: Inclusion criteria should include recent Pneumocystis prophylaxis, with a statistical analysis to discard a potential confounding variable.

Response: Thanks for the comment. We forgot to state that the patients were either not on treatment or poorly complied to it. We corrected the text as follows (lines 101 – 102):

“…the presence of subacute cough and dyspnea (≥ 7 days), a current TCD4 count < 250 cells/mm³, and poor compliance to or not on ART and PcP prophylaxis.”

Comment: Line 108 – now line 101: According to the CDC Revised Surveillance Case Definition for HIV Infection (2014), for stage 3 disease, the CD4 T cell count should be <200 cells/mm3; please provide your reference on <250 cells.

Response: In fact, most patients with PcP have less than 200 CD4 T cells counts. However, PcP also occurs in patients with slightly more than 200 CD4 T cells, albeit more rarely; e.g., 10% to 15% of the PcP episodes can occur in patient with CD4 T cells > 200, as reported below:

                        Phair, J et al. The Risk of Pneumocystis Carinii Pneumonia among Men Infected with Human Immunodeficiency Virus Type 1. N. Engl. J. Med. 1990, 322, 161–165, doi:10.1056/NEJM199001183220304.

                        Huang, L.; et al, Pneumocystis Workshop Participants An Official ATS Workshop Summary: Recent Advances and Future Directions in Pneumocystis Pneumonia (PCP). Proc Am Thorac Soc 2006, 3, 655–664, doi:10.1513/pats.200602-015MS.

Therefore, we chose CD4 T cell counts < 250 cells/mL as admission criterium because we also wanted to capture patients with colonization, in addition to those with PcP. In fact, we enrolled only 5 patients with suspected PcP and CD4>200 cells/mL: One was diagnosed as PcP, one as colonization and three as uninfected. 

Comment: Line 162-164: If sputum samples are variable, results might not be trusted. Although you addressed this issue in the Discussion section, there should be a way to minimize the variation in the sample collection, for example, quantifying total protein content, or performing a statistical analysis that show no difference between samples.

Response: The collection was performed under supervision of one of the authors (OJC) and ensured that a minimum of 15 minutes of inhalation was undertaken in each patient. Besides it, it was required that at least 1 ml of the sample was sent to DNA extraction.

There are methods to ensure the quality of the sputum, such as overall cellularity count. Induced sputum guarantees access to lung peripheral areas, as discussed by Pitchenik AE, et al. Sputum examination for the diagnosis of Pneumocystis carinii pneumonia in the acquired immunodeficiency syndrome. Am Rev Respir Dis 1986;133(2):226-9.

Since we were not able to stain all samples with O toluidine blue, we were also not able to perform overall cellularity analyses. This limitation is stated in lines 400 - 404: “Microscopic examinations were not available for most patients enrolled in our study due to the lack of specialized personnel. These methods usually require laboratory technicians with specific training. They are usually employed as the gold standard, especially when BALF samples are used; with IS their sensitivity tends to decrease.”

.

Comment: Line 169 – now 163: Could you explain why you chose 150 pg/mL as the B-glucan cutoff? Other studies have a higher cutoff value (see doi: 10.2478/raon-2020-0028)

Response: There is a debate in the literature on higher BDG values for discriminating PcP, which varied from 100 to 400 pg/mL [e.g., refs below]:

            Damiani, C et al. Combined Quantification of Pulmonary Pneumocystis Jirovecii DNA and Serum (1→3)-β- d -Glucan for Differential Diagnosis of Pneumocystis Pneumonia and Pneumocystis Colonization. J Clin Microbiol 2013, 51, 3380–3388, doi:10.1128/JCM.01554-13.

            Juniper, T et al. Use of β-D-Glucan in Diagnosis of Suspected Pneumocystis Jirovecii Pneumonia in Adults with HIV Infection. Int J STD AIDS 2021, 095646242110222, doi:10.1177/09564624211022247.

We thank the reviewer for suggestion of the interesting article, which escaped from our review. We included this reference in the revised version. (Line 370). We opted for a value of 150 pg/mL because when we crossed the data between BDG and Cq values, although our qPCR+ patients in general presented a low Cq (median 22.8), 9 patients had a BDG ranging between 150 and 400 pg/mL. In fact, if we had considered a cutoff of 400 pg/mL, these 9 patients would have been missed. Thus, according to our results, a cutoff value of 150 pg/mL was the most suitable to distinguish patients with PcP.

This manuscript is a resubmission of an earlier submission. The following is a list of the peer review reports and author responses from that submission.

Round 1

Reviewer 1 Report

The authors present an interesting study that tries to answer a specific need for clinicians: distinguishing between Pneumocystis disease and colonization, specially in patients living with HIV.

Author Response

We would like to thank the reviewer for the careful reading of our article and for the nice comments.

Point 1: Line 33-34: The qPCR development aims to distinguish Pneumocystis colonization from disease. Could you explain how this development enhances the knowledge about the natural history of the infection?

Response 1: We still know little about the natural history of the P. jirovecii infection in humans, owing to our difficulty in isolating the fungus, in obtaining respiratory samples, etc., but also because of our difficulty in identifying colonized individuals. We think that once we are able to more easily diagnose and follow these individuals, we will better understand its natural history

Point 2: Line 46: “Pneumocystis” should be in italics.

Response 2: Thanks, we corrected the text.

Point 3: Line 47: Nowadays, the term “antiretroviral treatment (ART)” is widely used instead of highly active antiretroviral treatment (HAART).

Response 3: Thanks, we corrected the text.

Point 4: Line 60-62 and line 66-69: This comment on the tests’ cost might need an additional cost-benefit analysis reflecting the overall cost, including equipment or laboratory expenses, which surely depend on the country they are acquired.

Response 4: Our claim that the qPCR assay is relatively low cost compared with commercial immunofluorescence and BDG assays, takes into account that, nowadays even in developing countries. laboratories that perform molecular diagnosis are more easily available. Thus, the cost of implementing a qPCR assay in these labs is relatively low when purchase and installation of PCR platforms are not considered.

Point 5: Line 77 and further: According to UNAIDS (2015), “aids” should be written in small letters.

Response 5: Thanks, we corrected the text.

Point 6: Line 99-101: This statement would fit better as conclusion.

Response 6: Thanks for the comment. We moved the statement to the conclusion of the article, as follows: “In conclusion, this study shows that our in-house qPCR targeting the mtSSU gene from IS samples was able to discriminate three groups among aids patients according to the Cq values: one with PcP, one without PcP, and a third one with colonized patients.’

We finished the introduction with a more concise and attenuated information on our qPCR.

Point 7: Line 107: Inclusion criteria should include recent Pneumocystis prophylaxis, with a statistical analysis to discard a potential confounding variable.

Response 7: Thanks for the comment. We forgot to state that the patients were either not on treatment or poorly complied to it also regarding PcP prophylaxis. We corrected the text as follows: “the presence of subacute cough and dyspnea (≥ 7 days), a current TCD4 count < 250 cells/mm³, and poor compliance to or not on ART and PcP prophylaxis.” (Lines 117 of the revised version)

Point 8: Line 108: According to the CDC Revised Surveillance Case Definition for HIV Infection (2014), for stage 3 disease, the CD4 T cell count should be <200 cells/mm3; please provide your reference on <250 cells.

Response 8: In fact, most patients with PcP have less than 200 CD4 T cells counts. However, PcP also occurs in patients with slightly more than 200 CD4 T cells, albeit more rarely; according to the references below, 10% to 15% of the PcP episodes can occur in patient with > 200 CD4 T cell counts.

Phair, J. et al. The Risk of Pneumocystis Carinii Pneumonia among Men Infected with Human Immunodeficiency Virus Type 1. Multicenter AIDS Cohort Study Group. N Engl J Med 1990, 322, 161–165, doi:10.1056/NEJM199001183220304.

Huang, L. et al. ATS Pneumocystis Workshop: Recent Advances and Future Directions in Pneumocystis Pneumonia (PCP). Proc Am Thorac Soc 2006, 3, 655–664, doi:10.1513/pats.200602-015MS.

Therefore, we choose CD4 T cell counts < 250 cells/mL as admission criterium because we wanted to also capture patients with colonization, in addition to PcP. In fact, only 5 patients with suspected PcP and CD4>200 cells were enrolled: one of them was diagnosed with PcP, one with colonization and 3 were uninfected. 

Point 9: Line 157-160: Microorganism names should be in italics; some of the genera are abbreviated and others, complete. Please, check the spelling of K. pneumoniae.

Response 9: Thanks, we corrected the text.

Point 10: Line 162-164: If sputum samples are variable, results might not be trusted. Although you addressed this issue in the Discussion section, there should be a way to minimize the variation in the sample collection, for example, quantifying total protein content, or performing a statistical analysis that show no difference between samples.

Response 10: The collection was performed under supervision of one of the authors (OJC) to ensure that a minimum of 15 minutes of inhalation was undertaken in each patient. Besides it, it was required that at least 1 ml of the sample was sent to DNA extraction.

There are analyzing methods to ensure the quality of the sputum, as overall cellularity count. Induced sputum guarantees access to lung peripherals areas, as presented below.

Pitchenik AE, Ganjei P, Torres A, Evans DA, Rubin E, Baier H. Sputum examination for the diagnosis of Pneumocystis carinii pneumonia in the acquired immunodeficiency syndrome. Am Rev Respir Dis 1986;133(2):226-9.

Since we were not able to staine all samples with O toluidine blue, we were also not able to perform overall cellularity analyses, implying a limitation of our study.

Point 11: Line 169: Could you explain why you chose 150 pg/mL as the B-glucan cutoff? Other studies have a higher cutoff value (see doi: 10.2478/raon-2020-0028)

Response 11: There is a concern on higher BDG values for discriminating PcP, varying in the literature from 100 to 400 pg/mL [e.g., refs below].

               Damiani, C.; Le Gal, S.; Da Costa, C.; Virmaux, M.; Nevez, G.; Totet, A. Combined Quantification of Pulmonary Pneumocystis Jirovecii DNA and Serum (1→3)-β- d -Glucan for Differential Diagnosis of Pneumocystis Pneumonia and Pneumocystis Colonization. J Clin Microbiol 2013, 51, 3380–3388, doi:10.1128/JCM.01554-13.

               Juniper, T.; Eades, C.P.; Gil, E.; Fodder, H.; Quinn, K.; Morris-Jones, S.; Gorton, R.L.; Wey, E.Q.; Post, F.A.; Miller, R.F. Use of β-D-Glucan in Diagnosis of Suspected Pneumocystis Jirovecii Pneumonia in Adults with HIV Infection. Int J STD AIDS 2021, 095646242110222, doi:10.1177/09564624211022247.

We thank for calling our attention to this interesting article, which went unnoticed in our review. It is now cited in the revised version (line 374, reference 81).

We opted for a value of 150 pg/mL since crossing over the results of the BDG assay and Cq values, we noted that our positive patients presented a median Cq of 22.84, including 9 patients which BDG were between 150 and 400 pg/mL. In fact, if we considered a cutoff of 400 pg/mL, 9 patients were be missed. So according to our results, a cutoff value of 150 pg/mL were suitable in distinguishing patients with PcP.

Point 12: It would be more understandable to read section 3.6 (Clinical and laboratory findings) after section 3.2 (patients enrolled).

Response 12: Thanks for the suggestion. However, as we used the PCR results in the section 3.6 to separate the groups for further clinical, laboratory and radiology characterization, we opted to keep it after the PCR results were presented. We hope that the reviewer will agree with this decision.

Point 13: Regarding p-values in Table 1, could you explain how was the comparison between groups performed? And which comparison does each p-value belong to? Also, it would be interesting to see the values from the statistical analysis in figures 2 to 4.

Response 13: Continuous variables were analyzed using ANOVA adjusted for white, and categorical variables were analyzed using chi-square. We provided the p value when there was a significant difference between two groups in the revised Table 1. We hope it is clearer now.

On the contrary, with regard to the data on the figures, their interpretation was illustrative to visualize the differing distribution of Cq values between the patients’ groups, and no statistical analysis is applicable.

Point 14: Line 404: This last statement wipes out the interesting finding you have enhanced through the whole paper, that your in-house qPCR is a great diagnostic tool. I agree that further investigation including more and different PcP patients is needed, but I suggest expressing more confidence on your test

Response 14: Thanks for this statement. We rewrote the final conclusion as follows: “In conclusion, this study shows that our in-house qPCR targeting the mtSSU gene from IS samples was able to discriminate three groups among aids patients according to the Cq values: one with PcP, one without PcP, and a third one with colonized patients.”

Reviewer 2 Report

The authors propose a qPCR assay in non-invasive sputum samples to improve diagnosis of PcP in patients living with HIV.  They compare a new qPCR with conventional PCR results.  They claim (line 97 and after) their assay has "very good performance", and that discriminates between colonized and PCP patients. Is this sufficient to guide clinical management as claimed?

The mansucript is too long and can be more concisely written by improving the focus. The Introduction section is excesive and diffuse and can be reduced to a half. A receiving operating characteristic (ROC) curve analysis with sensitivity, specificity, and predictive values will provide greater strength and validate the assay. The comparator in AIDS is normally microscopy or the conventional PCR. Predictive values are needed to document the superiority of qPCR over conventional PCR. 

Could the "high clinical index" be also, a comparator for this analysis?

How was sputum quality evaluated? (line 108).

Was microscopy in sputum done? 

Line 138: Please check the name of the primers. Both are listed as pAZ102-E.

Figure 1 can be ommited and indicated in the methods section of the manuscript. 

Author Response

We would like to thank the reviewer for the careful reading of our article and for the nice comments.

Point 1: The authors propose a qPCR assay in non-invasive sputum samples to improve diagnosis of PcP in patients living with HIV.  They compare a new qPCR with conventional PCR results.  They claim (line 97 and after) their assay has "very good performance", and that discriminates between colonized and PCP patients. Is this sufficient to guide clinical management as claimed?

Response 1: Thanks for the comment. We replaced “very good” by “high”, which we believe is more appropriate. (Line 108)

With this qPCR we were able to discriminate PcP from non-infected and colonized in our setting, assuring that anti-PcP treatment should be prescribed in aids patients with a positive qPCR Cq value < 31.0, and not in patients with a qPCR Cq value > 33.0. Regarding colonized patients, we propose that “we propose that our colonized group (31 ≥ Cq < 33) should not start immediately on anti-PcP treatment but be closely monitored since it is known that colonization can eventually lead to the development of PcP” (Lines 406-408).

Point 2: The manuscript is too long and can be more concisely written by improving the focus. The Introduction section is excessive and diffuse and can be reduced to a half.

Response 2: We reduced the introduction to half of the size of the original submission. We hope it is more focused now.

Point 3: A receiving operating characteristic (ROC) curve analysis with sensitivity, specificity, and predictive values will provide greater strength and validate the assay. The comparator in AIDS is normally microscopy or the conventional PCR. Predictive values are needed to document the superiority of qPCR over conventional PCR. Could the "high clinical index" be also, a comparator for this analysis?

Response 3: There are many previous papers proposing molecular assays for the diagnosis of PcP in immunosuppressed patients. In fact, in several of them the predictive values and/or ROC curve were calculated. However, they were all calculated based on comparators that are not appropriate since with low accuracy, potentially missing a significant number of infected patients. The high clinical index also suffers from the same limitation: in our cohort it missed patients with PcP and included patients without PcP. In general, these studies concluded that one single assay was not sufficient for the diagnosis of PCP, and most studies did not reach the routine clinical practice

In our study, the qualitative and quantitative PCR showed 100% agreement when patients with PcP and patients not infected were considered, but discrepant results in colonized patients as discussed in lines 250-258 of the revised version. We think that calculating predictive values or a ROC curve, in addition to the mentioned comparators limitations, would not improve our analyses or interpretation of the results. 

Point 4: How was sputum quality evaluated? (line 108).

Response 4: Sputum collection was performed under supervision of one of the authors (OJC) to ensure that a minimum of 15 minutes of inhalation was undertaken for every patient. Appropriately collected induced sputum allows access to secretion from lung peripherals areas, as discussed in the reference below. In addition, our protocol mandated that one mL of the IS samples was used for the DNA extractions. This was added to the Mat&Met section.

Pitchenik AE et al. Sputum examination for the diagnosis of Pneumocystis carinii pneumonia in the acquired immunodeficiency syndrome. Am Rev Respir Dis 1986;133(2):226-9.

Methods to evaluate the quality of the sputum are usually based on overall cellularity count. Although microscopic examination of sputum was planned initially, technical problems led to the interruption of this routine and we were unable to carry out cellularity analysis, a limitation we acknowledge might have occurred in an occasional patient (e.g., false negative patient #37 and eventually patient #30) as discussed in lines 380-387).

Point 5: Was microscopy in sputum done? 

Response 5: Sputum microscopy was done in the patients first enrolled, but it was unfortunately discontinued. This is commented in lines 129 – 132 of the Mat&Met section.

Point 6: Line 138: Please check the name of the primers. Both are listed as pAZ102-E.

Response 6: Thanks for the observation. The names were corrected.

Point 7: Figure 1 can be omitted and indicated in the methods section of the manuscript. 

Response 7: Figure 1 was removed.  The information is already indicated in the Mat&Met section. 

Round 2

Reviewer 2 Report

The authors improved their manuscript that is now more clear. The introduction still can ommit some background information to focus in the validation of a new test for diagnosis. 

Unfortunately the lack of controls and microscopy diagnosis makes it not possible to compare the sensitivity and specificity of the assay to known parameters. Therefore, although the work is interesting, the assay validation is not yet sufficient. 

The clinical index is an excellent contribution to the work presented.